# Confirming the statistically significant superiority of tree-based machine learning algorithms over their counterparts for tabular data

**Shahadat Uddin** [ORCID] *, **Haohui Lu**

School of Project Management, Faculty of Engineering, The University of Sydney, Forest Lodge, NSW, Australia

* shahadat.uddin@sydney.edu.au

**Data Availability Statement:** The 200 datasets used in this study are publicly available from open-source repositories. S2 Table contains the web address of each dataset.

## Abstract

Many individual studies in the literature observed the superiority of tree-based machine learning (ML) algorithms. However, the current body of literature lacks statistical validation of this superiority. This study addresses this gap by employing five ML algorithms on 200 open-access datasets from a wide range of research contexts to statistically confirm the superiority of tree-based ML algorithms over their counterparts. Specifically, it examines two tree-based ML (Decision tree and Random forest) and three non-tree-based ML (Support vector machine, Logistic regression and k-nearest neighbour) algorithms. Results from paired-sample t-tests show that both tree-based ML algorithms reveal better performance than each non-tree-based ML algorithm for the four ML performance measures (accuracy, precision, recall and F1 score) considered in this study, each at $p<0.001$ significance level. This performance superiority is consistent across both the model development and test phases. This study also used paired-sample t-tests for the subsets of the research datasets from disease prediction (66) and university-ranking (50) research contexts for further validation. The observed superiority of the tree-based ML algorithms remains valid for these subsets. Tree-based ML algorithms significantly outperformed non-tree-based algorithms for these two research contexts for all four performance measures. We discuss the research implications of these findings in detail in this article.

## 1. Introduction

Machine learning (ML) algorithms harness various statistical, probabilistic, and optimisation techniques to extrapolate historical data and discern valuable insights from vast, intricate, unstructured datasets [1]. ML has shown significant promise across various studies, including advancements in semantic embeddings [2], unsupervised learning techniques [3–5], disease prediction [6–8] and visual recognition optimisations [9–11]. Among the ML models, tree-based algorithms have gained prominence for their effectiveness, particularly in dealing with tabular data. These algorithms, including Decision Trees (DT) and Random Forests (RF),

**Funding:** The author(s) received no specific funding for this work.

**Competing interests:** The authors have declared that no competing interests exist.

operate through a hierarchical structure that enables transparent, criteria-based decision-making, adeptly managing both categorical and continuous inputs [12]. Such an inherent structure allows these models to compartmentalise the predictor space into simple regions, proving especially advantageous when addressing data embodying complex, non-linear relationships.

Tree-based algorithms partition the feature space into distinct and mutually exclusive regions. It forecasts outcomes using a series of test conditions arranged hierarchically. Each node in this structure evaluates a feature value, directing input from its root to the terminal leaves [13]. The ultimate prediction is typically derived from the dominant class or the average forecast of the samples within that leaf. This foundational approach in tree-based methods allows for capturing subtle data patterns, striking a harmonious balance between model robustness and explanatory clarity. On the other hand, non-tree-based algorithms such as Logistic regression (LR) and Support vector machine (SVM) adopt distinct methodologies for classification tasks. For instance, LR, a popular supervised learning classification algorithm developed in the 1940s, differentiates from linear regression by using a binomial output and employing the natural logarithm of the odds for its response variable to produce continuous criteria [14]. Further, SVM can address non-linear relationships using kernel methods. They might not consistently offer the level of interpretability found in tree-based models [15].

The choice between tree-based and non-tree-based algorithms depends on various considerations, such as the type of data, the importance of model clarity, and the need for generalisation. Given the wide-ranging usage of ML algorithms across different fields, it is essential to grasp the merits and drawbacks of each method. In this study, we conduct an in-depth comparison of tree-based algorithms with non-tree-based methods across various tabular datasets, aiming to highlight where tree-based algorithms excel.

Although studies in the current literature empirically showed the superiority of tree-based ML algorithms, no study shows such superiority through a statistical significance test. Following a classical comparative statistical significance test (paired-sample t-test), this study will show the performance superiority of tree-based ML algorithms over non-tree-based ML algorithms. We will use four ML performance measures (accuracy, precision, recall and F1 score) for this.

## 2. Literature review

Tabular data has long been the cornerstone of data analytics. Before the tree-based algorithms, k-nearest neighbour (KNN), LR and SVM were the standard choices for processing tabular data. While these methods excel in certain situations, they frequently struggle with non-linear data patterns and complex feature interplay [12].

Tree-based models like DT and RF have since filled this gap. They adeptly capture non-linear relationships and intricate data patterns by hierarchically partitioning the feature space [13]. DT is an interpretable model that can manage both numerical and categorical data. Further, RF is an ensemble method that boosts prediction accuracy and combats overfitting by averaging multiple decision tree outcomes. The advantages of tree-based models are numerous. For example, they naturally handle feature interactions, eliminating the need for tedious feature engineering [13]. Also, their interpretability has been enhanced by techniques, such as SHAP values, ensuring that complex models remain transparent [16].

In the literature, evidence from many benchmarking individual studies provides proof of the superiority of tree-based models. Perlich et al. [17] compared tree-based methods and logistic regression, evaluating their classification accuracy and the quality of rankings based on class membership probabilities. Their findings highlighted the superiority of the tree-based model. Later, Caruana and Niculescu-Mizil [18] compared different ML methods on 30

datasets. Tree-based models consistently outperformed non-tree-based algorithms regarding eight performance metrics. Although they illustrated empirical evidence of the superiority of tree-based ML algorithms, they did not show any statistical significance evidence, either at $p \leq 0.05$ *or* $p \leq 0.1$ levels, for their findings.

Like Caruana and Niculescu-Mizil [18], Fernández-Delgado et al. [19] experienced the same empirical evidence. They evaluated 179 classifiers from 17 families across various platforms, using 121 datasets primarily from the UCI database and some proprietary real-world problems. The results indicate that RF are the most likely top performers. Recently, deep learning has gained popularity. Yet, its dominance over tabular data remains debatable, even though it has shown success with text and image datasets. Uddin et al. [20] analysed the findings from 48 studies. They found that SVM is the most used ML algorithm, and RF is the one showing the best accuracy at most times. Grinsztajn et al. [21] compared tree-based models such as XGBoost and RF across 45 diverse datasets. The results revealed that tree-based models outperformed deep learning in general, especially on medium-sized data sets. Tree-based models excel on tabular data due to their inductive biases, while deep learning struggles with irregular target patterns and rotation invariance, especially when dealing with extraneous features in tabular datasets.

Many studies in the current literature empirically demonstrated the superiority of tree-based ML algorithms. They primarily used one or more datasets for descriptive statistical comparisons [e.g., 22]. Yet, employing statistical significance comparisons like t-tests to demonstrate such supremacy is not widespread. This study analyses the performance of five ML algorithms (two tree-based and three non-tree-based) over 200 datasets to demonstrate the superiority of tree-based algorithms over their counterparts at a statistical significance level.

## 3. Materials and methods

### 3.1 Data source

This study uses 200 open-access tabular datasets from the UCI Machine Learning Repository (53) and Kaggle (147). These two repositories host various research datasets that researchers can access for their research studies without any ethical obligation [23, 24]. The latter one hosts more datasets than the former one. The 200 datasets that this study considered are from 132 unique sources (S2 Table). Some of these sources contain more than one dataset. For example, we evaluated 50 datasets on university-ranking data from the Ultimate University Ranking source [25], containing ranking data from eight ranking-producing organisations, including QS and Times Higher Education, for several years. These 200 datasets are from various research contexts (Fig 1), including disease prediction (66), university- ranking (50), sports (23), finance (15) and academia (14). Acknowledging the potential for selection bias, we have carefully considered the diversity and representativeness of the datasets concerning the research questions addressed in this study. Our study meticulously selected a diverse range of datasets from various domains while ensuring a balance in dataset sizes and types to robustly mitigate the impact of selection bias on our research findings. For example, we opted for mean-based imputation to address missing data in numerical attributes within unskewed datasets [26]. For categorical data, we followed the mode-based imputation. This study followed the statistical approach of the synthetic minority oversampling technique [27] to make an unbalanced dataset a balanced one.

### 3.2 Machine learning algorithms

This study considers five ML algorithms balancing tree-based and non-tree-based approaches to harness their complementary strengths in addressing our research question.

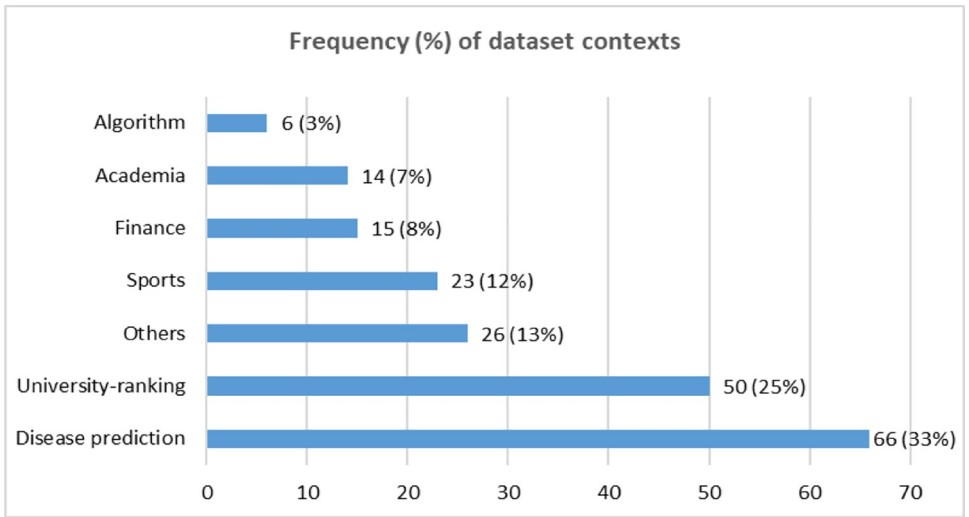

**Fig 1. Frequency and percentage of different dataset contexts.** The percentage figure within the bracket indicates the corresponding percentage value of the left number.

## Tree-based

Tree-based ML algorithms can map non-linear relationships well, making them more adaptable in solving classification and regression problems. They use several if-then conditional rules to develop prediction models. The study used two tree-based ML algorithms. They are Decision tree (DT) and Random Forest (RF). A DT, resembling a natural tree, is a hierarchical tree-like model consisting of multiple levels. In each level, different conditions are being tested and based on these test outcomes, the algorithm either reaches a final decision or jumps to a test condition of the next level [28]. A DT primarily consists of decision nodes and leaves. A sub-node which further divided into multiple sub-nodes is called a decision node. A sub-node is called a leaf node when it does not further split into additional sub-nodes. Leaf nodes contain the final decision outcome. RF is a commonly used ML algorithm that combines outputs from multiple DTs to reach a single result. Like the fact that a forest has many trees, an RF consists of several DTs [13]. Depending on the underlying problem, the determination of the results will vary. For a classification task, the majority voting will yield the final predicted categorical outcome. Outcomes from individual DTs are averaged for a regression task. These algorithms allow us to model complex interactions effectively, providing a solid basis for comparison against linear approaches.

## Non-tree-based

The three non-tree-based ML algorithms considered in this study are Support vector machine (SVM), Logistic regression (LR) and K-nearest neighbours (KNN). SVM is the most used supervised ML algorithm. An SVM operates a decision boundary, known as the hyperplane, for classification [29]. Data points on either side of this boundary line belong to different classes. Data points closer to the hyperplane on both sides are called support vectors. These points define the orientation and positioning of the hyperplane. SVM often uses kernel functions to handle the non-linearity of the decision surface for classification. LR estimates the probability of an event occurring on a scale between 0 and 1 [30]. Therefore, we must set a threshold value to use LR for a binary classification task. For example, a value ≤0.5 for a data instance will classify it as 'class A'; otherwise, 'class B'. LR can also be used for problems with more than two

classes through its generalised version, multinomial LR. KNN seeks votes from its *k* nearest neighbours to determine the class of a new data instance [31]. The class suggested by most of these votes will be the class of that new instance. There are many algorithms to quantify the nearest neighbours, including Euclidian distance and cosine similarity. The selection of these algorithms allows us to explore a range of decision boundaries from linear to non-linear, assessing their performance and interpretability in the context of our specific research question.

By considering both tree-based and non-tree-based algorithms, this study aims to comprehensively evaluate ML strategies, ensuring a robust and versatile analysis that can navigate the varied landscape of our research challenges. This deliberate choice of algorithms facilitates a nuanced understanding of different ML approaches, their strengths and limitations, enabling us to offer more grounded and practical insights into their applicability and performance.

### 3.3 Confusion matrix and performance measure

A confusion matrix, also known as an error matrix, is a tabular tool to demonstrate the performance of a classification algorithm [32]. Values in a confusion matrix can be of four types (Fig 2). True-positive (TP) is when a model correctly predicts a positive class. Similarly, True-negative (TN) is when a model correctly predicts a negative class. False-positive and False-negative happen when a model incorrectly predicts a positive and negative class, respectively.

Following the approaches followed in previous studies [e.g., 33, 34], this study uses four performance measures based on these four confusion matrix values. They are accuracy, precision, recall and F1 score. Here are their eqs.

$$Accuracy = \frac{TP + TN}{TP + TN + FP + FN}$$

$$Precision = \frac{TP}{TP + FP}$$

**Fig 2. Confusion matrix.**

$$Recall = \frac{TP}{TP + FN}$$

$$F1 = \frac{2 \times Precision \times Recall}{Precision + Recall}$$

### 3.4 Paired-sample t-test

The paired sample t-test, also known as the dependent sample t-test, is a statistical method used to check whether the mean difference between two observed groups is statistically significant [35]. Below is the formula for the paired sample t-test.

$$t = \frac{\bar{d}}{\frac{s}{\sqrt{n}}}$$

Where $\bar{d}$ and $s$ are the mean and standard deviation of all pairwise difference values, and $n$ is the total number of pairs in the dataset.

### 3.5 Experimental setup

This study used the Scikit-learn library [36] to implement the five ML algorithms with each research dataset considered in this study. Each dataset underwent an 80:20 split for the training and test data separation, and the training model development followed a five-fold cross-validation. To promote reproducibility, we adhered to Scikit-learn default parameters for all algorithms, ensuring a transparent and standardised experimental framework. While we applied the preprocessing steps uniformly across datasets, these were limited to essential procedures like normalisation, handled intrinsically by Scikit-learn [36]. For the paired sample t-test, we used IBM SPSS Statistics, version 28.0.0.0 [35], with its default setup for this test.

## 4. Results

Table 1 presents the paired-sample t-test results between the trained tree-based and non-tree-based ML algorithms for the four performance measures we used in this study. As illustrated in the *mean* column of this table, tree-based DT and RF show much higher values than the non-tree-based SVM, LR and KNN for all four performance measures. The differences are at $p<0.001$ significant level for each performance measure. Notably, the mean difference between two tree-based ML algorithms is minimal for each performance measure, leading to identical t-values in many cases.

Table 2 represents the number of times (out of 200 datasets) each ML algorithm performed best during the training phase concerning the four performance metrics when applied to the 200 research datasets considered in this study. Tree-based RF and DT outperformed the three non-tree-based ones by a significant margin. Interestingly, non-tree-based SVM, LR and KNN have the same score against all four performance measures. The row sum for each row in this table is higher than 200 since, in many cases, multiple ML algorithms performed best for the same measure against the same dataset. For example, DT showed the best accuracy score for all 200 datasets. RF also revealed the same best accuracy score for 194 datasets. For SVM, it is only 17.

Of the 200 datasets this study considered for research investigation, 66 are from the disease prediction context. The second largest is the university-ranking context, which has 50 datasets. We employed the paired-sample t-tests on the datasets of these two contexts for further in-

**Table 1. Paired-sample t-test results for the four performance measures between trained tree-based and non-tree-based supervised machine learning algorithms for the 200 datasets considered in this study.**

| Test | Group details | | Mean | | STD | | N | t | Sig. |
|---|---|---|---|---|---|---|---|---|---|
| | Tree-based | Non-tree-based | Mean 1 | Mean 2 | Std 1 | Std 2 | | | |
| (a) Accuracy | | | | | | | | | |
| 1 | Random forest | Support vector machine | 0.99838 | 0.92889 | 0.89796 | 8.57100 | 200 | 11.67 | <0.001 |
| 2 | Random forest | Logistic regression | 0.99838 | 0.90277 | 0.89796 | 10.64105 | 200 | 12.86 | <0.001 |
| 3 | Random forest | K-nearest neighbour | 0.99838 | 0.91919 | 0.89796 | 8.28808 | 200 | 13.77 | <0.001 |
| 4 | Decision tree | Support vector machine | 0.99839 | 0.92889 | 0.89801 | 8.57100 | 200 | 11.67 | <0.001 |
| 5 | Decision tree | Logistic regression | 0.99839 | 0.90277 | 0.89801 | 10.64105 | 200 | 12.86 | <0.001 |
| 6 | Decision tree | K-nearest neighbour | 0.99839 | 0.91919 | 0.89801 | 8.28808 | 200 | 13.77 | <0.001 |
| (b) Precision | | | | | | | | | |
| 1 | Random forest | Support vector machine | 0.99843 | 0.92166 | 0.86829 | 10.28040 | 200 | 10.74 | <0.001 |
| 2 | Random forest | Logistic regression | 0.99843 | 0.89307 | 0.86829 | 12.27288 | 200 | 12.29 | <0.001 |
| 3 | Random forest | K-nearest neighbour | 0.99843 | 0.91798 | 0.86829 | 8.39003 | 200 | 13.82 | <0.001 |
| 4 | Decision tree | Support vector machine | 0.99845 | 0.92166 | 0.85806 | 10.28040 | 200 | 10.74 | <0.001 |
| 5 | Decision tree | Logistic regression | 0.99845 | 0.89307 | 0.85806 | 12.27288 | 200 | 12.30 | <0.001 |
| 6 | Decision tree | K-nearest neighbour | 0.99845 | 0.91798 | 0.85806 | 8.39003 | 200 | 13.82 | <0.001 |
| (c) Recall | | | | | | | | | |
| 1 | Random forest | Support vector machine | 0.99839 | 0.92889 | 0.89796 | 8.57087 | 200 | 11.67 | <0.001 |
| 2 | Random forest | Logistic regression | 0.99839 | 0.90277 | 0.89796 | 10.64105 | 200 | 12.86 | <0.001 |
| 3 | Random forest | K-nearest neighbour | 0.99839 | 0.91910 | 0.89796 | 8.28808 | 200 | 13.77 | <0.001 |
| 4 | Decision tree | Support vector machine | 0.99840 | 0.92889 | 0.89801 | 8.57087 | 200 | 11.67 | <0.001 |
| 5 | Decision tree | Logistic regression | 0.99840 | 0.90277 | 0.89801 | 10.64105 | 200 | 12.86 | <0.001 |
| 6 | Decision tree | K-nearest neighbour | 0.99840 | 0.91910 | 0.89801 | 8.28808 | 200 | 13.77 | <0.001 |
| (d) F1 score | | | | | | | | | |
| 1 | Random forest | Support vector machine | 0.99838 | 0.91907 | 0.90220 | 10.21913 | 200 | 11.16 | <0.001 |
| 2 | Random forest | Logistic regression | 0.99838 | 0.89171 | 0.90220 | 12.15084 | 200 | 12.57 | <0.001 |
| 3 | Random forest | K-nearest neighbour | 0.99838 | 0.91569 | 0.90220 | 8.62228 | 200 | 13.82 | <0.001 |
| 4 | Decision tree | Support vector machine | 0.99836 | 0.91907 | 0.92129 | 10.21913 | 200 | 11.17 | <0.001 |
| 5 | Decision tree | Logistic regression | 0.99836 | 0.89171 | 0.92129 | 12.15084 | 200 | 12.58 | <0.001 |
| 6 | Decision tree | K-nearest neighbour | 0.99836 | 0.91569 | 0.92129 | 8.62228 | 200 | 13.83 | <0.001 |

depth investigation and validation. Table 3 details the corresponding findings only for the accuracy measure. S1(A)–S1(F) Table outlines the results for the precision, recall and F1 score measures. DT and RF have the exact mean accuracy and recall value for the disease prediction datasets. For the university-ranking context, they have the same mean value for all four performance measures. These two tables echo the findings from Tables 1 and 2; tree-based ML algorithms revealed superior performance compared to non-tree-based ones at $p<0.001$ significant level for the datasets of these two research contexts. It is worth noting that the tree-

**Table 2. Frequency statistics of the best performance of different trained machine learning algorithms against four performance metrics.**

| | Tree-based | | Non-tree-based | | |
|---|---|---|---|---|---|
| | Random forest | Decision tree | Support vector machine | Logistic regression | K-nearest neighbours |
| Accuracy | 194 | 200 | 17 | 13 | 9 |
| Precision | 188 | 198 | 17 | 13 | 9 |
| Recall | 194 | 200 | 17 | 13 | 9 |
| F1 score | 193 | 193 | 17 | 13 | 9 |

**Table 3. Paired-sample t-test results for the accuracy measure between tree-based and non-tree-based supervised machine learning algorithms for the datasets from disease prediction and university-ranking contexts.**

| Test | Group details | | Mean | | STD | | N | t | Sig. |
|---|---|---|---|---|---|---|---|---|---|
| | Tree-based | Non-tree-based | Mean 1 | Mean 2 | Std 1 | Std 2 | | | |
| (a) Disease prediction context (66 datasets) | | | | | | | | | |
| 1 | Random forest | Support vector machine | 0.99575 | 0.89766 | 1.47404 | 10.02376 | 66 | 8.093 | <0.001 |
| 2 | Random forest | Logistic regression | 0.99575 | 0.87265 | 1.47404 | 11.62626 | 66 | 8.703 | <0.001 |
| 3 | Random forest | K-nearest neighbour | 0.99575 | 0.89919 | 1.47404 | 9.30741 | 66 | 8.630 | <0.001 |
| 4 | Decision tree | Support vector machine | 0.99575 | 0.89766 | 1.47404 | 10.02376 | 66 | 8.093 | <0.001 |
| 5 | Decision tree | Logistic regression | 0.99575 | 0.87265 | 1.47404 | 11.62626 | 66 | 8.703 | <0.001 |
| 6 | Decision tree | K-nearest neighbour | 0.99575 | 0.89919 | 1.47404 | 9.30741 | 66 | 8.630 | <0.001 |
| (b) University-ranking context (50 datasets) | | | | | | | | | |
| 1 | Random forest | Support vector machine | 1.00000 | 0.99189 | 0.00000 | 1.10923 | 50 | 5.171 | <0.001 |
| 2 | Random forest | Logistic regression | 1.00000 | 0.98652 | 0.00000 | 1.56146 | 50 | 6.105 | <0.001 |
| 3 | Random forest | K-nearest neighbour | 1.00000 | 0.97987 | 0.00000 | 2.07197 | 50 | 6.871 | <0.001 |
| 4 | Decision tree | Support vector machine | 1.00000 | 0.99189 | 0.00000 | 1.10923 | 50 | 5.171 | <0.001 |
| 5 | Decision tree | Logistic regression | 1.00000 | 0.98652 | 0.00000 | 1.56146 | 50 | 6.105 | <0.001 |
| 6 | Decision tree | K-nearest neighbour | 1.00000 | 0.97987 | 0.00000 | 2.07197 | 50 | 6.871 | <0.001 |

based ML algorithms attained a possible highest score against all four performance measures for the university ranking datasets.

To ensure validation, we conducted t-tests between tree-based and non-tree-based ML algorithms for their performance obtained during the test phase. The outcomes of these t-tests resembled those that resulted when comparing their performance during the training phase (Table 1). Table 4 presents an instance of such results for the accuracy measure. This study observed similar findings from the test phase for the other three performance measures (precision, recall and F1 score).

## 5. Discussion

This study uses statistical tests to compare the performance between tree-based ML and non-tree-based algorithms. The results consistently indicate that tree-based ML algorithms outperform their non-tree-based counterparts across all four performance measures, with the differences being statistically significant at the $p < 0.001$ level. This superiority remains consistent when delving deeper into specific datasets, especially those related to disease prediction and university ranking. Tree-based algorithms attained an impeccable score for the university-ranking datasets, hinting at their effectiveness in this domain. The exceptional accuracy scores

**Table 4. Paired-sample t-test results for the accuracy measure between tree-based and non-tree-based supervised machine learning algorithms during the test phase for the 200 datasets considered in this study.**

| Test | Group details | | Mean | | STD | | N | t | Sig. |
|---|---|---|---|---|---|---|---|---|---|
| | Tree-based | Non-tree-based | Mean 1 | Mean 2 | Std 1 | Std 2 | | | |
| 1 | Random forest | Support vector machine | 0.92455 | 0.89635 | 0.10654 | 0.13134 | 200 | 6.150 | <0.001 |
| 2 | Random forest | Logistic regression | 0.92455 | 0.89065 | 0.10654 | 0.13665 | 200 | 4.491 | <0.001 |
| 3 | Random forest | K-nearest neighbour | 0.92455 | 0.88230 | 0.10654 | 0.12493 | 200 | 5.914 | <0.001 |
| 4 | Decision tree | Support vector machine | 0.90969 | 0.89635 | 0.12295 | 0.13134 | 200 | 2.637 | 0.005 |
| 5 | Decision tree | Logistic regression | 0.90969 | 0.89065 | 0.12295 | 0.13665 | 200 | 2.354 | 0.100 |
| 6 | Decision tree | K-nearest neighbour | 0.90969 | 0.88230 | 0.12295 | 0.12493 | 200 | 3.445 | <0.001 |

achieved in our models prompt considerations of overfitting. However, through stringent cross-validation, we have ensured the robustness and generalisability of our results. Furthermore, the evident distinctions among classification groups identified in our analysis greatly enhance the discernibility, affirming the prediction accuracy of the models considered in the study within the specific attributes of the underlying dataset.

There could be several reasons for the superior performance of the tree-based ML algorithms compared with their counterparts. The most notable one is that, unlike linear models, they can map non-linear relations very well, empowering them with excellent prediction accuracy and greater stability [37]. Moreover, they can better accommodate categorical and numerical data than others [38]. Tree-based ML algorithms can be described as sets of *if-else* statements, enabling them to assimilate non-linear and categorical data during the model learning process. This results in enhanced predictive accuracy.

Datasets from Kaggle and UCI Machine Learning Repository often exhibit high prediction accuracy. The 50 University-ranking datasets show 100% prediction accuracy, as detailed in Table 3(b). There are several reasons behind the presence of such high accuracy. Datasets are carefully preprocessed, cleaned, and removed inconsistencies, missing values and outliers before making them available on these two open-access platforms. They may also have well-engineered features that simplify the modelling process and enhance prediction performance. For these reasons, models often developed and tested based on these open-access datasets reveal superior accuracy. However, caution should be taken for their real-world applications that often encounter diverse, unclean and complex data.

The implications of our findings are manifold. Confirming the superiority of the tree-based ML algorithms will help future researchers make appropriate data analysis plans for their studies. For classification, some studies in the literature [e.g., 39] employed only the non-tree-based ML algorithms on tabular data. Uddin et al. [20] pointed out that SVM is the most used ML algorithm in the disease prediction literature. Our findings suggest that, along with other ML algorithms, researchers should consider at least one tree-based one for addressing a classification task using a tabular dataset. This suggestion offers a clear indication towards the choice of algorithm, especially when tackling tabular datasets.

One of the limitations of this study is that it considered only classical tree-based and non-tree-based supervised ML algorithms for comparison. It did not evaluate other ML algorithms, such as deep learning ones. Although RF is one of the classical ML algorithms, it is an ensemble approach. Our study did not consider other practical ensemble approaches. For example, this study did not consider tree-based AdaBoost and XGBoost ensemble ML algorithms like other studies [e.g., 40] in the literature. Similarly, we did not consider unsupervised ML algorithms such as k-means clustering for performance comparison. Another notable limitation of this study is that although it statistically demonstrated the superiority of tree-based ML algorithms, it did not explore the underlying reasoning behind such dominance. It is evident in the literature that tree-based ML algorithms can handle non-linear classification datasets better [13], which could be a possible reason for their performance superiority. Uddin and Lu [41] noticed that dataset meta-level and statistical attributes do not impact the performance of tree-based MLs. However, they have a statistically significant impact on non-tree-based ML algorithms. Further, we recognise that our focus on standard performance metrics like accuracy, precision, recall, and F1 score may not fully capture the complexity of model evaluation, omitting metrics such as Area under the receiver operating characteristic curve and specificity. Future studies will aim to incorporate a broader array of metrics, ensuring a more comprehensive assessment of classification model performance in alignment with research objectives.

These limitations could help define new future research scopes and opportunities. Exploring the performance of ensemble tree-based algorithms with the results of this study can offer

more comprehensive insights. While our findings suggest tree-based algorithms outperform non-tree-based ones across multiple datasets, we recognise the importance of considering dataset-specific characteristics, such as feature distribution and complexity, that could influence algorithm performance. Uddin and Lu [41] discovered that ML algorithms exhibit varying performances when applied to datasets with distinct meta-level and statistical attributes. Moreover, an explanatory approach, combined with domain expertise, could unearth the factors contributing to the superiority of tree-based algorithms. Such insights can refine algorithmic choices and expand our understanding of the underlying mechanics of these algorithms. To further enrich the field, future studies should investigate the efficacy of deep learning (e.g., convolutional neural networks), unsupervised learning (e.g., k-means clustering) and ensemble tree-based algorithms (e.g., AdaBoost and XGBoost), and dissect the factors influencing algorithmic performance disparities. Detailed comparative analyses and examining feature importance across varied datasets will be crucial in these endeavours, offering a more straightforward path toward optimising ML applications in various domains.

## 6. Conclusion

Non-tree-based ML algorithms performed inferior to tree-based ones at a statistically significant level. Many individual studies in the current literature also pointed out this kind of superiority of tree-based ones over non-tree-based ones. To our knowledge, this study is the first one that confirmed such supremacy statistically by employing two tree-based and three non-tree-based classical ML algorithms on 200 datasets from various research contexts. Future studies can consider other tree-based and non-tree-based ML algorithms (e.g., ensemble ones) to explore such dominance of the former ones using research datasets from different contexts. Until then, our findings can provide insightful details in selecting appropriate ML algorithms for future researchers to design their research analyses and experimental setups.

## Supporting information

**S1 Table. Paired-sample t-test results for the precision, recall and F1 score measures between tree-based and non-tree-based trained supervised machine learning algorithms for the datasets from disease prediction (66) and university-ranking contexts (50).** (DOCX)

**S2 Table. Dataset source information.** (DOCX)

## Acknowledgments

We acknowledge the University of Sydney's Vacation Research Internship recipient, Palak Mahajan, for her contribution to dataset extraction and preprocessing.

## Author Contributions

**Conceptualization:** Shahadat Uddin.

**Data curation:** Shahadat Uddin.

**Formal analysis:** Shahadat Uddin, Haohui Lu.

**Methodology:** Shahadat Uddin.

**Software:** Shahadat Uddin.

**Supervision:** Shahadat Uddin.

**Writing – original draft:** Shahadat Uddin, Haohui Lu.

**Writing – review & editing:** Shahadat Uddin, Haohui Lu.

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
