## [Decision Letter · Decision Letter 0]

22 Feb 2024

PONE-D-24-03825Confirming the statistically significant superiority of tree-based machine learning algorithms over their counterparts for tabular dataPLOS ONE

Dear Dr. Uddin,

Thank you for submitting your manuscript to PLOS ONE. After careful consideration, we feel that it has merit but does not fully meet PLOS ONE’s publication criteria as it currently stands. Therefore, we invite you to submit a revised version of the manuscript that addresses the points raised during the review process.

We look forward to receiving your revised manuscript.

Kind regards,

Nagarajan Raju

Academic Editor

PLOS ONE

Journal Requirements:

Additional Editor Comments (if provided):

I suggest authors to go through the reviewers comments and address them properly in the revised manuscript.

Reviewers' comments:

Reviewer's Responses to Questions

**Comments to the Author**

1. Is the manuscript technically sound, and do the data support the conclusions?

Reviewer #1: No

Reviewer #2: Yes

Reviewer #3: No

Reviewer #4: Yes

2. Has the statistical analysis been performed appropriately and rigorously? 

Reviewer #1: I Don't Know

Reviewer #2: Yes

Reviewer #3: No

Reviewer #4: Yes

3. Have the authors made all data underlying the findings in their manuscript fully available?

Reviewer #1: No

Reviewer #2: Yes

Reviewer #3: No

Reviewer #4: Yes

4. Is the manuscript presented in an intelligible fashion and written in standard English?

Reviewer #1: No

Reviewer #2: Yes

Reviewer #3: No

Reviewer #4: Yes

5. Review Comments to the Author

Reviewer #1: 1. The study relies on tabular datasets from the UCI Machine Learning Repository and Kaggle, which are popular repositories for machine learning datasets. However, there's a risk of selection bias as these datasets may not be representative of real-world data or may have biases inherent in their collection process.

2. The study chooses five machine learning algorithms, but the rationale for selecting these specific algorithms is not thoroughly justified. While tree-based and non-tree-based algorithms are commonly used, there should be a discussion on why these particular algorithms were chosen over others and how they complement each other in addressing the research question.

3. While the study uses common performance metrics such as accuracy, precision, recall, and F1 score, there's limited discussion on why these metrics were chosen and how they align with the research objectives. Additionally, there's no mention of other important metrics such as area under the receiver operating characteristic curve (AUC-ROC) or specificity, which are crucial for evaluating classification models.

4. The study mentions using the Scikit-learn library for implementing machine learning algorithms and IBM SPSS Statistics for conducting paired-sample t-tests. While these are widely used tools, the lack of detailed information on specific parameter settings and preprocessing techniques could hinder reproducibility. Providing a clear and detailed description of the experimental setup would enhance the study's transparency and reproducibility.

5. The study splits the data into training and test sets using an 80:20 ratio and performs five-fold cross-validation during model development. While cross-validation helps assess the model's performance, there's no external validation using independent datasets.

6. The study focuses solely on classical tree-based and non-tree-based supervised ML algorithms, neglecting other important techniques such as deep learning algorithms or unsupervised learning algorithms.

7. Measurement metrics (i.e., accuracy, recall, etc.) are well-known and have been used in previous biomedical studies such as PMID: 36642410, PMID: 28155651. Therefore, the authors are suggested to refer to more works in this description to attract a broader readership.

8. The study does not consider ensemble approaches beyond Random Forest (RF), such as AdaBoost or XGBoost, which have shown significant performance improvements in various classification tasks.

9. While the study demonstrates the superiority of tree-based ML algorithms, it fails to explore the underlying reasons behind this dominance.

10. The study suggests that tree-based algorithms consistently outperform non-tree-based algorithms across all datasets, without considering potential dataset-specific factors that may influence algorithm performance.

11. While the study briefly mentions future research opportunities, such as exploring ensemble tree-based algorithms and investigating the underlying reasons for algorithmic performance, it lacks depth in discussing specific research avenues and methodologies.

Reviewer #2: My Comments are as follow:

1) The study focuses on classical algorithms and may not include recent advancements in machine learning, such as deep learning techniques that have shown promise in handling tabular data. It is highly recommended to include these for more contemporary perspective.

2) Abstract does not highlight novelty of the proposed work. It’s better to add more specific details of your work.

3) Introduction is not focused and literature can be reorganised to strengthen literature review following contributions and discuss few relevant works i.e.,

a) A Benchmark Dataset and Learning High-level Semantic Embeddings of Multimedia for Cross-media Retrieval

b) Unsupervised pre-trained filter learning approach for efficient convolution neural network

c) CSFL: A novel unsupervised convolution neural network approach for visual pattern classification

d) Optimization of CNN through novel training strategy for visual classification problems

e) Face recognition: A novel un-supervised convolutional neural network method

f) ModPSO-CNN: an evolutionary convolution neural network with application to visual recognition

g) Two-stage domain adaptation for infrared ship target segmentation

4) The work does not delve deeply into the impact of feature engineering and data preprocessing steps, which are crucial for the performance of machine learning algorithms. Add a detail discussion on it.

5) While the proposed work effectively compares tree-based algorithms with non-tree-based counterparts, it might lack a deeper analysis of why certain algorithms perform better than others. A more thorough investigation into the intrinsic properties of the datasets that favour tree-based methods is needed.

Reviewer #3: The paper is not scentifically sound to be published in this form.

Reviewer #4: The study aims to investigate the statistical significance of the performance of decision tree-based algorithms over other classical machine learning algorithms. Some points need modification in a final version. The manuscript's idea is interesting, since it seems inappropriate for articles on machine learning algorithms not to conduct statistical comparisons between the accuracies obtained by these algorithms in classification tasks.

Abstract and Introduction

-"no study has shown such supremacy through a statistical significance test." and "However, none shows such supremacy by employing any statistical significance comparison, such as a t-test." It's not true; below I can indicate an example that used statistics to compare the accuracy of machine learning algorithms, and it is possible that others have proceeded similarly. I suggest the authors rewrite the sentence and indicate that it is not usual to find statistical comparisons between the classification performance of machine learning algorithms.

Farias, F. M., Salomão, R. C., Rocha Santos, E. G., Sousa Caires, A., Sampaio, G. S. A., Rosa, A. A. M., Costa, M. F., & Silva Souza, G. (2023). Sex-related difference in the retinal structure of young adults: a machine learning approach. Frontiers in medicine, 10, 1275308. https://doi.org/10.3389/fmed.2023.1275308.

Methods

-Figure 1: Use a dot instead of a comma for decimal numbers. Include the label name for the X-axis.

-It would be important to provide more information about the type of data used. Time series for subsequent feature extraction? Was feature extraction performed? If yes, how many and which ones were extracted? Were they the same for all comparisons? How many groups were used in different datasets?

-Why was the t-test chosen over an analysis of variance? I think it would be more appropriate to use an analysis of variance or Kruskal-Wallis or perform a Bonferroni correction for the t-test results.

-I suggest performing at least a 10-fold cross-validation.

-Was there data preprocessing? Any normalization? I think it would be important.

-Does it make sense to compare the performance of random forest and decision tree?

Results

- Indicate the standard deviation of the mean values in Table 1 and Table 3.

- Table 3 shows accuracy of 1. Does it imply overfitting? Or do the groups exhibit very large differences, leading to easier classification? This debate could be done in Discussion section

6. PLOS authors have the option to publish the peer review history of their article (what does this mean?). If published, this will include your full peer review and any attached files.

Reviewer #1: No

Reviewer #2: No

Reviewer #3: No

Reviewer #4: No

---

## [Author Response · Author response to Decision Letter 0]

8 Mar 2024

Reviewer response

Confirming the statistically significant superiority of tree-based machine learning algorithms over their counterparts for tabular data

We sincerely thank the reviewers and editor for their insightful suggestions and comments. Here is our response to the corrections suggested by each of them. Changes are marked in red colour in the revised main manuscript file.

Suggestions from the Editor

Comment 1: 

Please ensure that your manuscript meets PLOS ONE’s style requirements, including those for file naming. The PLOS ONE style templates can be found at 

Our response: Thank you very much for this suggestion. We paid particular attention to meeting the PLOS ONE style requirements while revising this manuscript.

Comment 2: 

I suggest authors go through the reviewers' comments and address them adequately in the revised manuscript.

Our response: Thank you for this suggestion. We gave our best effort to address all comments by the reviewers adequately.

Reviewer: 01

Comment 1: The study relies on tabular datasets from the UCI Machine Learning Repository and Kaggle, which are popular repositories for machine learning datasets. However, there is a risk of selection bias as these datasets may not represent real-world data or may have inherent biases in their collection process.

Our response: Thank you for pointing out this selection bias issue. We have taken steps to avoid this kind of bias as much as possible. The selected datasets considered in our study are from a wide range of contexts, as outlined in Figure 1 (page 6). In addition, we followed different statistical approaches, such as mean-based imputation [1] for handling the missing data problem and the synthetic minority oversampling technique [2] to make an unbalanced dataset a balanced one. Please see lines 140-147 on page 5 for further information.

Comment 2: The study chooses five machine learning algorithms, but the rationale for selecting these specific algorithms is not thoroughly justified. While tree-based and non-tree-based algorithms are commonly used, there should be a discussion on why these particular algorithms were chosen over others and how they complement each other in addressing the research question.

Our response: We appreciate this comment. Accordingly, we have revised Section 3.2 to clarify our rationale for selecting specific tree-based and non-tree-based algorithms, highlighting their complementary strengths in addressing our research question. This revision ensures a balanced exploration of machine learning strategies, enhancing the methodological rigour of our study and the relevance of our findings. Please see lines 161-162 (page 6), 177 –179 (page 6), and 194 – 201 (page 7) for further details. 

Comment 3: While the study uses standard performance metrics such as accuracy, precision, recall, and F1 score, there is limited discussion on why these metrics were chosen and how they align with the research objectives. Additionally, there is no mention of other important metrics, such as area under the receiver operating characteristic curve (AUC-ROC) or specificity, which are crucial for evaluating classification models.

Our response: Thank you for highlighting the importance of diverse evaluation metrics in addition to the commonly used four we considered (i.e., accuracy, precision, recall and F1 score). In response, we have acknowledged this limitation in our manuscript and emphasised our intention to explore additional metrics such as AUC-ROC and specificity in future research to provide a more holistic evaluation of model performance. We added these to the limitations and future study scope of our study. Please see lines 368 – 372 on page 14.

Comment 4: The study mentions using the Scikit-learn library to implement machine learning algorithms and IBM SPSS Statistics to conduct paired-sample t-tests. While these are widely used, the lack of detailed information on specific parameter settings and preprocessing techniques hinders reproducibility. A clear and precise description of the experimental setup would enhance transparency and reproducibility.

Our response: Thank you for emphasising the importance of detailing our experimental setup. We have clarified in the manuscript that we used default Scikit-learn and IBM SPSS settings for all algorithms and statistical tests to ensure straightforward reproducibility. For the t-test, we used the SPSS default parameter setting for this test. Our approach aims for maximum transparency and ease of replication. Please check lines 238 – 243 on page 8 for further details. 

Comment 5: The study splits the data into training and test sets using an 80:20 ratio and performs five-fold cross-validation during model development. While cross-validation helps assess the model performance, there is no external validation using independent datasets.

Our response: We acknowledge your feedback. In this revised edition, we have incorporated external validation. To achieve this, we applied the five ML algorithms considered in our study to the test dataset, which was not previously encountered during the training phase. The results we obtained were consistent with those from the training phase (Table 1). Additionally, we have introduced a new table (Table 4 on page 12) illustrating specific outcomes from the new t-tests, focusing solely on the accuracy measure. The other three measures (precision, recall and F1 score) showed similar superiority for the tree-based ML algorithms. For further details, please refer to lines 306-310 on page 12.

Comment 6: The study focuses solely on classical tree-based and non-tree-based supervised ML algorithms, neglecting other essential techniques such as deep learning algorithms or unsupervised learning algorithms.

Our response: Thank you for highlighting the exclusive focus of our study on classical tree-based and non-tree-based supervised machine learning algorithms. The decision to concentrate on these algorithms was deliberate, rooted in our research's specific scope and objectives, which aimed to investigate and compare the effectiveness of traditional ML approaches in our domain. While we acknowledge the potential of deep learning, ensemble ML algorithms and unsupervised ML algorithms in advancing the field, we intentionally leave them as a potential future scope. Please see pages 382-387) on pages 14-15 for more information.

Comment 7:

Measurement metrics (i.e., accuracy, recall, etc.) are well-known and have been used in previous biomedical studies, such as in PMID: 36642410 and PMID: 28155651. Therefore, the authors are suggested to refer to more works in this description to attract a broader readership.

Our response: Thank you for suggesting additional seminal works on measurement metrics. We included references to critical studies to underscore the relevance of our chosen metrics in biomedical research and broaden the manuscript's appeal. Please see line 209 on page 7 for details.

Comment 8: The study does not consider ensemble approaches beyond Random Forest (RF), such as AdaBoost or XGBoost, which have significantly improved performance in various classification tasks.

Our response: We appreciate this comment. The second reviewer also made a similar comment (R2C1). In the revised manuscript, we have discussed this. We also mentioned that deep learning could have a chance of showing such superior performance. We leave them as potential future research scopes in alignment with our study. Please see lines 359-360 on page 13 for details. We also added further studies could delve into these methods. Please see lines 382 – 387 for details. 

Comment 9: While the study demonstrates the superiority of tree-based ML algorithms, it fails to explore the underlying reasons behind this dominance.

Our response: We have discussed the possible underlying reasons behind the superiority of tree-based ML algorithms compared to their counterparts. The superior performance of tree-based ML algorithms compared to their counterparts can be attributed to various factors. A prominent reason is their capability to effectively map non-linear relations, providing excellent prediction accuracy and greater stability, especially compared to linear models [3]. Additionally, these algorithms exhibit better categorical and numerical data accommodation than other models [4]. Described as a set of if-else statements, tree-based ML algorithms excel at incorporating non-linear and categorical data during the learning process, contributing to enhanced predictive accuracy. Please see lines 326-331 on page 13 for our detailed discussion.

Comment 10: The study suggests that tree-based algorithms consistently outperform non-tree-based algorithms across all datasets without considering potential dataset-specific factors that may influence algorithm performance.

Our response: Thank you for pointing out the importance of considering dataset-specific factors in evaluating algorithm performance. Acknowledging this, we have updated our discussion to more carefully examine how each dataset's unique attributes could influence the effectiveness of tree-based versus non-tree-based algorithms. Please refer to lines 375-379 in the revised manuscript for an expanded analysis, ensuring a more nuanced and balanced comparison.

Comment 11: While the study briefly mentions future research opportunities, such as exploring ensemble tree-based algorithms and investigating the underlying reasons for algorithmic performance, it lacks depth in discussing specific research avenues and methodologies.

Our response: We acknowledge the reviewer’s feedback on the need for a more detailed exploration of future research directions within our study. Future work will explore ensemble tree-based algorithms and the reasons behind varying algorithmic performances, employing comparative analyses and feature importance studies for deeper insights. We also added further studies for these. For further details, please see lines 382 – 387 on pages 14-15. 

Reviewer: 02

Comment 1: The study focuses on classical algorithms and may not include recent advancements in machine learning, such as deep learning techniques that have shown promise in handling tabular data. It is highly recommended to include these for a more contemporary perspective.

Our response: Thank you for this suggestion. The first reviewer also made a similar comment (R1C8). In the revised manuscript, we have discussed this. We also mentioned that ensemble approaches could have a chance of showing such superior performance. We leave them as potential future research scopes in alignment with our study. Please see lines 359-360 on page 13 for details. We also added further studies could delve into these methods. Please see lines 382 – 387 for more information.

Comment 2: The abstract does not highlight the novelty of the proposed work. It is better to add more specific details to your work.

Our response: We have revised the abstract considering our study aims and objectives. Please see page 2 for details.

Comment 3:

The introduction is not focused, and the literature can be reorganised to strengthen the literature review following contributions and discuss a few relevant works, i.e.,

(a) A Benchmark Dataset and Learning High-level Semantic Embeddings of Multimedia for Cross-media Retrieval

(b) Unsupervised pre-trained filter learning approach for efficient convolution neural network

(c) CSFL: A novel unsupervised convolution neural network approach for visual pattern classification

(d) Optimisation of CNN through novel training strategy for visual classification problems

(e) Face recognition: A novel un-supervised convolutional neural network method

(f) ModPSO-CNN: an evolutionary convolution neural network with application to visual recognition

(g) Two-stage domain adaptation for infrared ship target segmentation

Our response: Thank you for your constructive feedback. We have focused our introduction, reorganised the literature review to highlight significant contributions in CNN and machine learning advancements, and meticulously incorporated the suggested references, enhancing the clarity and depth of our study. Please see lines 54 – 56 on page 3. 

Comment 4: The work does not delve deeply into the impact of feature engineering and data preprocessing steps, which are crucial for the performance of machine learning algorithms. Add a detailed discussion on it.

Our response: Thank you for raising these issues related to feature engineering and data preprocessing. In this revised version, we briefly outlined the preprocessing steps followed for data analysis in this study. Please see lines 238-243 on page 8 for more information. To promote reproducibility, we adhered to Scikit-learn default parameters for all algorithms, ensuring a transparent and standardised experimental framework. 

 While our findings suggest tree-based algorithms outperform non-tree-based ones across multiple datasets, we recognise the importance of considering dataset-specific characteristics, such as feature distribution and complexity, that could influence algorithm performance. Uddin and Lu [5] discovered that ML algorithms exhibit varying performances when applied to datasets with distinct meta-level and statistical attributes. 

Comment 5: While the proposed work effectively compares tree-based algorithms with non-tree-based counterparts, it might lack a deeper analysis of why certain algorithms perform better than others. A more thorough investigation into the intrinsic properties of the datasets that favour tree-based methods is needed.

Our response: We appreciate your comment, which echoed the tenth comment from the first reviewer (R1C10), highlighting the necessity of acknowledging dataset-specific factors when assessing algorithm performance. In response, we have refined our discussion to more thoroughly explore the impact of each dataset's distinct characteristics on the performance of tree-based versus non-tree-based algorithms. For a detailed expansion of this analysis, which aims to offer a more nuanced and balanced comparison, please see lines 368-372 in the revised manuscript.

Reviewer: 03

Comment 1:

The paper is not scientifically sound to be published in this form.

Our response: We believe that the comments from the other three reviewers and our corresponding responses have significantly improved the scientific merit of this study. Incorporating these changes would make the revised manuscript scientifically rigorous for publication. Please see the revised manuscript for our responses concerning the comments made by the first, second and fourth reviewers.

Reviewer: 04

Comment 1:

Abstract and Introduction

-"no study has shown such supremacy through a statistical significance test." and "However, none shows such supremacy by employing any statistical significance comparison, such as a t-test." It's not true; below I can indicate an example that used statistics to compare the accuracy of machine learning algorithms, and it is possible that others have proceeded similarly. I suggest the authors rewrite the sentence and indicate that it is not usual to find statistical comparisons between the classification performance of machine learning algorithms.

Farias, F. M., Salomão, R. C., Rocha Santos, E. G., Sousa Caires, A., Sampaio, G. S. A., Rosa, A. A. M., Costa, M. F., & Silva Souza, G. (2023). Sex-related difference in the retinal structure of young adults: a machine learning approach. Frontiers in medicine, 10, 1275308. https://doi.org/10.3389/fmed.2023.1275308.

Our response: We appreciate for pointing out this issue. We have reviewed the mentioned article, which primarily followed descriptive statistics for ML performance comparison. We also found that many studies in the current literature empirically demonstrated the superiority of tree-based ML algorithms. They primarily used one or more datasets for descriptive statistical comparisons [e.g., 18]. Yet, employing statistical significance comparisons like t-tests to demonstrate such supremacy is not widespread. Please see lines 123-125 on page 5 for more information.

Comment 2:

Methods

-Figure 1: Use a dot instead of a comma for decimal numbers. Include the label name for the X-axis.

-It would be important to provide more information about the type of data used. Time series for subsequent feature extraction? Was feature extraction performed? If yes, how many and which ones were extracted? Were they the same for all comparisons? How many groups were used in different datasets?

-Why was the t-test chosen over an analysis of variance? I think it would be more appropriate to use an analysis of variance or Kruskal-Wallis or perform a Bonferroni correction for the t-test results.

-I suggest performing at least a 10-fold cross-validation.

-Was there data preprocessing? Any normalisation? I think it would be important.

-Does it make sense to compare the performance of random forest and decision tree?

Our response: Please see below for our responses against each point

- Figure 1: We intended to use a comma to show values both in raw value and its corresponding percentage. We have updated this figure in this revised submission. We put a bracket instead of a comma. We further updated the figure caption accordingly. Please see page 5 for more details.

- Our datasets are from a wide range of contexts. They have attributes ranging from 2 to 2,548. All these datasets have two groups for the target variable.

- Since we have only two groups for all datasets, we considered the independent sample t-test. ANOVA or Kruskal-Wallis is more suitable for datasets with more than two groups [6].

- We explore the size distribution for all 200 datasets to finalise the selection of 5-fold cross-validation. Some datasets are not large, and selecting a 10-fold cross-validation would lead to inappropriate results.

- The second reviewer also raised this point (R2C4). In this revised version, we briefly outlined the preprocessing steps followed for data analysis in this study. Please see lines 238-243 on page 8 for more information. To promote reproducibility, we adhered to Scikit-learn default parameters for all algorithms, ensuring a transparent and standardised experimental framework. While our findings suggest tree-based algorithms outperform non-tree-based ones across multiple datasets, we recognise the importance of considering dataset-specific characteristics, such as feature distribution and complexity, that could influence algorithm performance. Uddin and Lu [5] discovered that ML algorithms exhibit varying performances when applied to datasets with distinct meta-level and statistical attributes.

- We considered all classic supervised ML algorithms. Although RF is an ensemble approach based on DT, we considered it in our study in alignment with numerous studies in the literature.

Comment 3:

Results

- Indicate the standard deviation of the mean values in Table 1 and Table 3.

- Table 3 shows an accuracy of 1. Does it imply overfitting? Or do the groups exhibit very large differences, leading to easier classification? This debate could be discussed in the Discussion section

Our response: All relevant tables, including Tables 1 and 3, have been updated with the standard deviation values. We further update Supplementary Table 1 accordingly. Please see different tables for details. We have discussed the presence of such high accuracy in lines 339-347 on page 14.

Reference

1. Wei, R., Wang, J., Su, M., Jia, E., Chen, S., Chen, T., and Ni, Y., Missing value imputation approach for mass spectrometry-based metabolomics data. Scientific reports, 2018. 8(1): p. 663.

2. Ishaq, A., Sadiq, S., Umer, M., Ullah, S., Mirjalili, S., Rupapara, V., and Nappi, M., Improving the prediction of heart failure patients’ survival using SMOTE and effective data mining techniques. IEEE access, 2021. 9: p. 39707-39716.

3. Dumitrescu, E., Hué, S., Hurlin, C., and Tokpavi, S., Machine learning for credit scoring: Improving logistic regression with non-linear decision-tree effects. European Journal of Operational Research, 2022. 297(3): p. 1178-1192.

4. Song, Y.-Y. and Ying, L., Decision tree methods: applications for classification and prediction. Shanghai archives of psychiatry, 2015. 27(2): p. 130.

5. Uddin, S. and Lu, H., Dataset meta-level and statistical features affect machine learning performance. Scientific Reports, 2024. 14(1): p. 1670.

6. Field, A., Discovering statistics using SPSS. 2013, London: Sage Publications Ltd.

---

## [Decision Letter · Decision Letter 1]

18 Mar 2024

Confirming the statistically significant superiority of tree-based machine learning algorithms over their counterparts for tabular data

PONE-D-24-03825R1

Dear Dr. Uddin,

We’re pleased to inform you that your manuscript has been judged scientifically suitable for publication and will be formally accepted for publication once it meets all outstanding technical requirements.

Kind regards,

Nagarajan Raju

Academic Editor

PLOS ONE

Additional Editor Comments (optional):

Reviewers' comments:

Reviewer's Responses to Questions

**Comments to the Author**

1. If the authors have adequately addressed your comments raised in a previous round of review and you feel that this manuscript is now acceptable for publication, you may indicate that here to bypass the “Comments to the Author” section, enter your conflict of interest statement in the “Confidential to Editor” section, and submit your "Accept" recommendation.

Reviewer #1: All comments have been addressed

Reviewer #2: All comments have been addressed

Reviewer #4: All comments have been addressed

2. Is the manuscript technically sound, and do the data support the conclusions?

Reviewer #1: Yes

Reviewer #2: Yes

Reviewer #4: Yes

3. Has the statistical analysis been performed appropriately and rigorously? 

Reviewer #1: N/A

Reviewer #2: Yes

Reviewer #4: Yes

4. Have the authors made all data underlying the findings in their manuscript fully available?

Reviewer #1: No

Reviewer #2: Yes

Reviewer #4: Yes

5. Is the manuscript presented in an intelligible fashion and written in standard English?

Reviewer #1: Yes

Reviewer #2: Yes

Reviewer #4: Yes

6. Review Comments to the Author

Reviewer #1: My previous comments have been addressed, therefore, the manuscript can be accepted in this current form.

Reviewer #2: All my comments are successfully answered. Please take a good look to the grammar and typos while submitting the final version of the manuscript.

Reviewer #4: (No Response)

7. PLOS authors have the option to publish the peer review history of their article (what does this mean?). If published, this will include your full peer review and any attached files.

Reviewer #1: No

Reviewer #2: No

Reviewer #4: No

---

## [Editor Report · Acceptance letter]

3 Apr 2024

PONE-D-24-03825R1 

PLOS ONE

Dear Dr. Uddin, 

I'm pleased to inform you that your manuscript has been deemed suitable for publication in PLOS ONE. Congratulations! Your manuscript is now being handed over to our production team.

Kind regards, 

on behalf of

Dr. Nagarajan Raju 

Academic Editor

PLOS ONE